# Effectiveness of a Dental Intervention to Improve Oral Health among Home Care Recipients: A Randomized Controlled Trial

**DOI:** 10.3390/ijerph18179339

**Published:** 2021-09-03

**Authors:** Jonas Czwikla, Alexandra Herzberg, Sonja Kapp, Stephan Kloep, Heinz Rothgang, Ina Nitschke, Cornelius Haffner, Falk Hoffmann

**Affiliations:** 1Department of Health Services Research, Carl von Ossietzky University of Oldenburg, 26129 Oldenburg, Germany; falk.hoffmann@uni-oldenburg.de; 2Department of Health, Long-Term Care and Pensions, SOCIUM Research Center on Inequality and Social Policy, University of Bremen, 28359 Bremen, Germany; a.herzberg@uni-bremen.de (A.H.); sokapp@uni-bremen.de (S.K.); rothgang@uni-bremen.de (H.R.); 3High-Profile Area of Health Sciences, University of Bremen, 28359 Bremen, Germany; kloep@uni-bremen.de; 4Competence Center for Clinical Trials, University of Bremen, 28359 Bremen, Germany; 5Division of Gerodontology, Clinic of Prosthetic Dentistry and Dental Materials Science, University Medical Center, 04103 Leipzig, Germany; Ina.Nitschke@zzm.uzh.ch; 6Clinic of General, Special Care and Geriatric Dentistry, Center of Dental Medicine, University of Zurich, CH-8032 Zurich, Switzerland; 7Special Care- and Geriatric Dentistry, Städtisches Klinikum Harlaching München, 81545 Munich, Germany; haffner@teamwerk-deutschland.de

**Keywords:** geriatric dentistry, objective oral health, oral health-related quality of life, periodontitis, long-term care

## Abstract

We quantified the effectiveness of an oral health intervention among home care recipients. Seven German insurance funds invited home care recipients to participate in a two-arm randomized controlled trial. At t_0_, the treatment group (TG) received an intervention comprising an oral health assessment, dental treatment recommendations and oral health education. The control group (CG) received usual care. At t_1_, blinded observers assessed objective (Oral Health Assessment Tool (OHAT)) and subjective (Oral Health Impact Profile (OHIP)) oral health and the objective periodontal situation (Periodontal Screening Index (PSI)). Of 9656 invited individuals, 527 (5.5%) participated. In the TG, 164 of 259 (63.3%) participants received the intervention and 112 (43.2%) received an outcome assessment. In the CG, 137 of 268 (51.1%) participants received an outcome assessment. The OHAT mean score (2.83 vs. 3.31, *p* = 0.0665) and the OHIP mean score (8.92 vs. 7.99, *p* = 0.1884) did not differ significantly. The prevalence of any periodontal problems (77.1% vs. 92.0%, *p* = 0.0027) was significantly lower in the TG than in the CG, but the prevalence of periodontitis was not (35.4% vs. 44.6%, *p* = 0.1764). Future studies should investigate whether other recruitment strategies and a more comprehensive intervention might be more successful in improving oral health among home care recipients.

## 1. Introduction

Oral health is an important contributor to quality of life and wellbeing [1,2]. To maintain and improve oral health, proper oral hygiene and appropriate dental care are essential [3,4]. People in need of long-term care (LTC) are, however, typically less able to brush their own teeth, take care of their dentures and visit dental practices [5,6]. This can lead to a decline in oral health, the occurrence of dental pain, an increased risk of tooth loss, a deterioration of nutritional status and the development of systemic diseases [7,8,9].

Previous studies from Germany and France as well as a systematic review including studies from 19 countries around the world, found poor oral hygiene and health as well as a low utilization of dental care among both nursing home residents and home care recipients [10,11,12,13,14]. In many countries, existing interventions to maintain and improve the oral health status of people in need of LTC, however, focus primarily on the nursing home setting, which usually includes only up to a quarter of all LTC dependents, as the majority are home care recipients [15,16,17].

As in the nursing home setting, research from Germany, Sweden and the United States indicates that oral hygiene and utilization of dental care in home care recipients are often impeded by physical and mental disabilities [14,18,19]. In the home care setting, oral hygiene and dental care utilization can be neglected even further because formal caregivers trained in oral care are often not involved in the care process [20]. Moreover, home visits by dentists can be particularly time-consuming and are not widely available. Therefore, to improve the provision of dental care to community-dwelling LTC dependents, new forms of health care are urgently needed [21].

The objective of this study was to quantify the effectiveness of an oral health intervention comprising the provision of an oral health assessment, dental treatment recommendations and oral health education to home care recipients via a pro-active, low-threshold, outreach approach.

## 2. Materials and Methods

This two-arm randomized controlled trial (RCT) was conducted in cooperation with seven German statutory health and LTC insurance funds belonging to the BKK Dachverband. It was approved by the University of Bremen Ethics Committee (Ethics Committee Number: MundPflege; date of approval: 21 March 2018) and registered in the German Clinical Trials Register (Trial-ID: DRKS00013517). The trial included persons who were (i) a member of one of the seven cooperating insurance funds, (ii) aged ≥ 18 years, (iii) in need of LTC in accordance with the German Social Code Book XI (i.e., in need of permanent support to compensate physical and/or mental disabilities), (iv) in receipt of LTC benefits in the home care setting and (v) residing in the German federal states of Bremen or Lower Saxony.

The estimated total number of eligible people was 9500. Because all eligible persons (or their legal guardians) were invited to participate and approached for informed consent by the cooperating insurance funds, we expected to recruit approximately 1000 individuals (10.5% response). If more than 1000 persons had responded, we would have drawn up a waiting list. The invitation letter and one reminder were sent out by letter at the beginning and at the end of the second quarter of 2018, respectively. After providing informed consent, independent from the study team and insurance funds, the Competence Center for Clinical Trials of the University of Bremen assigned a sequential identification number to all participants. Stratified by insurance fund, the participants were then randomly assigned by the Competence Center for Clinical Trials to the treatment group (TG) or the control group (CG) with a 1:1 ratio. Block randomization was applied to ensure group balance. For this purpose, computer-generated random lists with a block length of six were used. All individuals were informed by letter about their group assignment.

To guide reporting, we followed the CONSORT (Consolidated Standards of Reporting Trials) statement [22] (Appendix A).

### 2.1. Intervention

Between May 2018 and November 2019 (t_0_), an oral health intervention was provided to the TG participants, while the CG participants received no intervention (dental care as usual). In Germany, usual dental care financed by the insurance funds includes, inter alia, dental prophylaxis, dental and periodontal treatment and the supply of dentures. Appointments for the provision of the intervention were scheduled by telephone by an appointment allocation service. The intervention was carried out in the domestic setting and comprised an oral health assessment, dental treatment recommendations and oral health education. The oral health assessment was conducted by one of the trained dentists and lasted between 20 and 30 min: First, the status of natural teeth, dentures and oral mucosa/tongue/gums were subjectively rated with “good”, “moderate”, or “poor”. Then, the need for dental treatment (response options “no”, “fillings”, “gums/mucosa”, “dental extraction”, “dentures” and “other”) and oral hygiene support (response options “no”, “partly” and “full”) were subjectively assessed. Finally, the dentist recommended dental treatment where necessary and coordinated the oral health education. Dental treatment was recommended to be carried out either at the patient’s home or at a dental practice. Oral health education was provided by the dentist during the same visit or his/her trained dental assistant during the same or during an additional visit in the domestic setting and also lasted between 20 and 30 min.

During the training, the participants received the following oral care tools: mouth rinse, toothpaste, toothbrushes, tongue cleaners and interdental brushes. Individuals with dentures also received denture adhesive and denture brushes. Built-up handles were provided if needed.

The dentists providing the intervention were recruited by the study team via digital newsletters of the Associations of Statutory Health Insurance Dentists of Bremen and Lower Saxony. The total number of dentists in Bremen and Lower Saxony was 7201 [23]. All these dentists were invited via newsletters. Additionally, a convenience sample of 195 dentists was contacted via personal correspondence. In total, 30 dentists declared their willingness to participate (0.4% response). The participating dentists were able to decide whether they wanted to provide the intervention at t_0_ (decided by 20 dentists) or assess the outcomes at t_1_ (decided by 10 dentists). All dentists and their dental assistants participating at t_0_ were trained by the German Society for Gerodontology and the study team using standardized training materials. After the project was introduced by the study team, two dentists of the German Society for Gerodontology provided education on oral health changes among older adults; associations between oral health and chronic diseases, multimorbidity and polypharmacy; tailored oral health education for people in need of LTC; and physical and mental limitations among LTC dependents. Finally, the study team provided training on the standardized provision of the intervention. All dentists participating at t_1_ were trained only by the study team. The training comprised an introduction to the project and education on the standardized assessment of the outcomes.

### 2.2. Outcome Assessment

Between January 2019 and November 2020 (t_1_), objective oral health (primary outcome), subjective oral health (secondary outcome I) and the objective periodontal situation (secondary outcome II) in the TG and CG were assessed either by a blinded trained dentist or—if no dentist was available (5% of all outcome assessments)—one of two blinded trained study nurses. In cases where a dentist was available, he/she assessed the primary and both secondary outcomes for a participant during one visit. In cases where only a study nurse was available, the secondary outcome II was not assessed to avoid adverse events. In both groups, the outcomes were only assessed at t_1_. The CG participants received no outcome assessment at t_0_, because it would have been unethical to assess dental care needs among these patients without subsequently recommending treatment. In addition, the latter might have resulted in a contamination of the CG. Consequently, the TG participants also received no outcome assessment at t_0_. The planned time between t_0_ and t_1_ was 6 months. Appointments for the outcome assessment were scheduled by telephone and assessments were carried out in the domestic setting. They lasted between 20 and 30 min among individuals with and approximately 15 min among individuals without natural teeth.

Objective oral health was assessed using the Oral Health Assessment Tool (OHAT), validated by Chalmers et al. [24]. OHAT includes eight categories (lips, tongue, gums and tissues, saliva, natural teeth, dentures, oral cleanliness and dental pain), each of which can be rated with “0 = healthy”, “1 = changes”, or “2 = unhealthy”. Since no German translation of the OHAT was available, we used the forward-backward translation method to translate the English version into German. A German translation has meanwhile been published [25] and is almost identical with our translation.

Subjective oral health (i.e., oral health-related quality of life) was assessed using a German version of the 14-item short-form Oral Health Impact Profile (OHIP), validated by John et al. [26]. OHIP comprises seven conceptual dimensions, each with two items (shown in parentheses): (i) functional limitation (trouble pronouncing words and taste worse), (ii) physical pain (painful aching and uncomfortable to eat), (iii) psychological discomfort (self-conscious and tense), (iv) physical disability (diet unsatisfactory and interrupt meals), (v) psychological disability (difficult to relax and been embarrassed), (vi) social disability (irritable with others and difficulty doing jobs) and (vii) handicap (life unsatisfying and unable to function). For each item, participants were asked how frequently they had experienced an oral health-related impact in the preceding month. Response options were “0 = never”, “1 = hardly never”, “2 = occasionally”, “3 = fairly often” and “4 = very often” [26,27].

The objective periodontal situation was assessed using the German Periodontal Screening Index (PSI), a widely used routine assessment applied in usual dental care and internationally known as “Periodontal Screening and Recording (PSR)” [28,29]. The PSI divides the jaw into sextants, that can be rated with “0 = no bleeding, no tartar or plaque”, “1 = bleeding, no tartar or plaque”, “2 = bleeding, tartar or plaque”, “3 = pocket depths 3.5–5.5 mm”, or “4 = pocket depths > 5.5 mm”.

### 2.3. Sample Size Calculation

Based on an OHAT mean score of 5.27 with a standard deviation (SD) of 2.10 and a standard normal distribution [30], as well as an alpha cut-off of 5% (α = 0.05) and an expected dropout of 25%, a sample size of 92 persons per group was needed for detecting a difference of one OHAT point (OHAT mean score of 4.27; d = 0.48) with a power of 80% (β = 0.20). Because we aimed to stratify our analysis by individuals receiving only informal care provided by relatives or friends (66.9% of all home care recipients) and individuals receiving also formal care provided by home care nursing services (33.1% of all home care recipients) [31], a sample size of at least 278 per group was originally planned.

### 2.4. Record Linkage

Sociodemographic data on sex and age as well as data on LTC grades and LTC benefits were obtained in the second quarter of 2018 for all participants from insurance claims data linked to primary data. LTC grades were originally assessed by the Medical Advisory Service and differentiate into five grades (higher LTC grades represent greater LTC dependency). The data on LTC benefits were originally assessed for billing purposes by the insurance funds and indicate whether the participants received LTC benefits by bank transfer only to organize informal care, or also in kind, i.e., formal care in the home care setting.

### 2.5. Statistical Analysis

First, the response of home care recipients was calculated by dividing the invited by the recruited number of home care recipients. The distributions of sex (male, female), age groups (<60, 60–74, 75–84, 85+ years), LTC grades (1/2 (low/substantial limitations), 3 (severe limitations), 4/5 (very severe limitations without/with special challenges for nursing care)) and type of LTC benefits (only informal care, also formal care) at t_0_ as well as the mean age at t_0_ were compared between TG and CG participants.

Second, the provision of the intervention at t_0_ was analyzed. The numbers of TG participants who utilized the different components of the intervention were determined. Furthermore, the proportions of TG participants with a poor status of natural teeth (only applicable to persons with natural teeth), dentures (only applicable to persons with dentures) and oral mucosa/tongue/gums at t_0_ were calculated. The proportions of TG participants with unmet dental care needs and those in need of oral hygiene support at t_0_ were also determined.

Finally, the effectiveness of the intervention was quantified. In both the TG and CG, the distributions of sex, age groups, LTC grades and type of LTC benefits as well as the mean age were compared between individuals whose primary outcomes were assessed/non-assessed at t_1_ using chi-square tests and the nonparametric Wilcoxon–Mann–Whitney-test. These comparisons were conducted to investigate whether the assessed participants differ from the dropouts. The same characteristics were also compared between the assessed TG and CG participants to examine potential bias resulting from differential dropouts. The means of the total OHAT scores for the TG and CG at t_1_ were compared using the Wilcoxon–Mann–Whitney-test. The proportions of the OHAT scores were compared per category and in total using chi-square and Fisher’s exact tests. Furthermore, a multivariable linear regression was conducted which considered the total OHAT score as the dependent variable, group (TG, CG) as the main explanatory variable and sex, age group, LTC grade, type of LTC benefits and time in days between randomization and t_1_ as control variables. The same procedures were applied to the OHIP scores. The prevalence of any periodontal problems and periodontitis in the TG and CG at t_1_ were compared using chi-square tests. Persons with any periodontal problems were identified using a dichotomized PSI variable (score 0 (or missing) for all sextants indicating no periodontal problems; score 1, 2, 3, or 4 for at least one sextant indicating periodontal problems). Individuals with periodontitis were identified using a further dichotomized PSI variable (score 0, 1, or 2 (or missing) for all sextants indicating no periodontitis; score 3 or 4 for at least one sextant indicating periodontitis) [28,29]. Two multivariable logistic regressions were also conducted. In these regressions, the prevalence of any periodontal problems and periodontitis, respectively, served as the dependent variable. The independent variables were the same as in the linear regressions.

All analyses were conducted using SAS 9.4 (SAS Institute Inc., Cary, NC, USA).

## 3. Results

### 3.1. Baseline Characteristics

The total number of eligible home care recipients was 9656. All of them were invited to participate and 527 declared their willingness to do so (5.5% response) (Figure 1). Of these, 259 were randomized into the TG and 268 into the CG. The proportion of women was 50.6% in the TG and 49.6% in the CG. The percentage distribution of age groups was 17.8% vs. 21.3% (<60), 20.5% vs. 20.1% (60–74), 38.6% vs. 34.7% (75–84) and 23.2% vs. 23.9% (85+ years). The mean age was 73.4 (SD: 16.7) and 71.5 (SD: 18.0) years, respectively. The percentage distribution of the LTC grades was 49.0% vs. 50.7% (grades 1/2), 29.3% vs. 28.4% (grade 3) and 21.6% vs. 20.9% (grades 4/5). The proportion of individuals receiving only informal care was 74.1% in the TG and 74.3% in the CG.

### 3.2. Intervention Provided at t_0_

The oral health intervention was provided to 164 (63.3%) of the 259 TG participants. The mean time between randomization and t_0_ was 159.3 days (SD: 166.2, median: 91.5). Oral health assessments were carried out on all 164 (100.0%) participants. Dental treatment was recommended to 107 (65.2%) of the assessed participants and oral health education was utilized by 153 (93.3%). Among 144 assessed persons with natural teeth, 19 (13.2%) had poor teeth. Of 118 assessed individuals with dentures, the dentures of 21 (17.8%) were in poor condition. Oral mucosa/tongue/gums were in poor health in 16 (9.8%) of the assessed persons. Unmet dental care needs were found among 102 (62.2%) and need of oral hygiene support among 55 (33.5%) participants.

### 3.3. Primary Outcome Assessed at t_1_

The primary outcome was assessed for 112 (43.2%) TG participants and 137 (51.1%) CG participants (Table 1). In the TG, the mean time between t_0_ and t_1_ was 337.9 days (SD: 141.6, median: 322.0). The mean time between randomization and t_1_ was 511.5 days (SD: 176.2, median: 488.5) in the TG and 425.1 days (SD: 204.4, median: 367.0) in the CG (*p* = 0.0001). In the TG, the distribution of sex, LTC grades and type of LTC benefits did not differ among assessed and non-assessed participants. However, the distribution of age groups differed and the mean age was lower among assessed participants than among non-assessed participants. In the CG, the distribution of sex and type of LTC benefits also did not differ between assessed and non-assessed participants. However, age as well as the proportions of participants with higher LTC grades (i.e., 4/5) were lower among assessed participants compared to non-assessed participants. The distribution of sex, age groups, LTC grades and type of LTC benefits as well as mean age did not differ between assessed TG and CG participants.

Regarding objective oral health, the total OHAT mean score did not differ significantly between the TG and CG (2.83 [SD 2.60] vs. 3.31 [SD 2.50], *p* = 0.0665). However, the percentage distribution of the total OHAT scores in the TG and CG differed: 22.3% vs. 11.0% (total score 0), 31.3% vs. 33.6% (total score 1–2) and 46.4% vs. 55.5% (total score 3+) (*p* = 0.0487). With regard to the individual OHAT categories, no differences were found (Table 2). In the linear regression, the total OHAT score was not significantly lower in the TG compared to the CG (−0.42 [95% confidence interval (CI) −1.09 to 0.25]; *p* = 0.2172).

The subgroup analysis indicated that the tendency for an oral health improvement was more pronounced among participants receiving informal care only compared to those also receiving formal care Appendix A.

### 3.4. Secondary Outcomes Assessed at t_1_

Regarding subjective oral health, the total OHIP mean score did not differ significantly between the TG and CG (8.92 [SD 9.86] vs. 7.99 [SD 10.55], *p* = 0.1884). Likewise, no difference was found for the percentage distribution of the total OHIP scores: 28.6% vs. 34.3% (total score 0–1), 22.3% vs. 26.3% (total score 2–5) and 49.1% vs. 39.4% (total score 6+) (*p* = 0.3081). With regard to the individual OHIP items, scores for “taste worse” were lower among TG participants than among CG participants, whereas scores for “difficulty doing jobs” were higher (Table 3). In the linear regression, also no difference was found (−0.13 [95% CI −2.79 to 2.54]; *p* = 0.9262).

In terms of the objective periodontal situation, the prevalence of any periodontal problems in the TG was significantly lower than in the CG (77.1% vs. 92.0%, *p* = 0.0027). The prevalence of periodontitis did not differ (35.4% vs. 44.6%, *p* = 0.1764). In the logistic regressions, the odds ratio for periodontal problems for TG vs. CG participants was 0.35 (95% CI 0.15 to 0.83; *p* = 0.0174), whereas the odds ratio for periodontitis was 1.00 (95% CI 0.53 to 1.88; *p* = 0.9955).

In the subgroup analysis, no subjective oral health differences and no difference regarding the prevalence of periodontitis between the TG and CG were found in both subgroups. The lower prevalence of periodontal problems was only observed among participants receiving only informal care (Appendix A).

## 4. Discussion

This RCT quantified the effectiveness of an oral health intervention comprising an oral health assessment, dental treatment recommendations and oral health education among home care recipients. The response of home care recipients was found to be poor and we did not reach the precalculated sample size and power. If provided, the intervention tended to improve objective oral health, but subjective oral health was not improved. Regarding the objective periodontal situation, the prevalence of any periodontal problems was reduced but not the prevalence of periodontitis.

With respect to reaching home care recipients for dental care provision in the domestic setting, our findings suggest that sending out invitation letters through insurance funds is unsatisfactory. To address low dental care utilization among home care recipients [14], alternative strategies for reaching this population group might be more successful. Studies from Germany, the Netherlands and Australia suggest that the involvement of general practitioners and formal caregivers already providing medical or nursing care to home care recipients could be a promising approach [32,33,34].

In our study, appointments for domiciliary dental care provision were successfully scheduled in two thirds of all cases. However, although the appointments were scheduled by an appointment allocation service, the limited number of participating dentists made it difficult to actually schedule appointments. To motivate more dentists to make home visits, those interested in geriatric dentistry should be equipped with mobile dental treatment facilities (e.g., mobile ultrasonic devises for removing tartar), which are currently, not widely available in many countries [5,32]. Furthermore, dentists should be adequately remunerated for the additional effort of providing domiciliary dental care [5,32]. Recent reforms in Germany, however, only marginally increased the remuneration for this type of dental care provision [6]. Finally, the implementation of centers for geriatric dentistry could help to ensure that an adequate number of dentists is available for the provision of dental care to people in need of LTC.

Moves to strengthen cooperation between general practitioners and dentists as well as between formal caregivers and dentists could help facilitate the coordination of domiciliary dental visits [32,35]. Moreover, research from the Netherlands and the United States suggest that a better integration of dental care into usual medical care would be worthwhile [36,37]. Recent reforms in Germany, however, have concentrated exclusively on strengthening cooperation between dentists and nursing homes, where treatment can be provided to multiple patients during one visit [6]. Currently, the German Network for Quality Development in Nursing is developing an expert standard on “Promoting Oral Health in Nursing”. For the first time, the development process involves dentists in addition to nurses [38].

With respect to the impact of our intervention, the tendency towards improved objective oral health and the amelioration of any periodontal problems at t_1_ among TG participants indicate that the combination of an oral health assessment, dental treatment recommendations and oral health education provided in the patients’ home improves oral health among home care recipients. The tendency for a poorer subjective oral health among TG participants at t_1_ might be explained by an increased awareness of personal oral health problems due to dentally assessed oral health problems during the provision of the intervention at t_0_. The non-significant change in the prevalence of periodontitis could be explained by the aspect that our intervention comprised no dental treatment and it was the decision of the participants whether to follow the treatment recommendations or not. As to the single effect of educational interventions provided in the LTC setting, the findings of previous systematic reviews including the international literature are inconsistent [17,39,40,41]. In their Cochrane Review, moreover, Albrecht et al. [17] emphasize that all studies conducted in the nursing home setting had a high or unclear risk of bias and neglected outcomes in oral health and oral health-related quality of life. More recent studies from Germany and Finland, however, indicate that educational interventions are conducive to improving oral health among people in need of LTC [42,43,44].

Overall, according to international studies, a combination of different measures appears to be the most promising approach to improving oral health among LTC dependents [45,46,47,48,49]. In Germany, taking general practitioners and formal caregivers on board in the coordination of dental care, as well as supporting dentists interested in providing dental treatment and oral health education in the patient’s domicile is essential [32]. The consideration of the special dental care needs of relevant subgroups of LTC dependents, such as people with cognitive impairments, would also be helpful as demonstrated in the international literature [50,51,52].

### Strengths and Limitations

One major strength of this study is that the effectiveness of an oral health intervention has been quantified in a RCT. Furthermore, the study was designed in such a way that no baseline assessment was needed in the CG. This enabled us to compare the intervention with current usual care. The pro-active, low-threshold, outreach approach of the intervention is a further strength of the study. All eligible individuals were invited to participate by the cooperating insurance funds (pro-active component), the appointments for the provision of the oral health intervention were scheduled by an appointment allocation service (low-threshold component) and the intervention was provided in the domestic setting (outreach component). This allowed us to address the prevailing access barriers that impede the utilization of dental care among home care recipients.

There are, however, some important limitations. First, the response among home care recipients to the invitation letters was lower than expected (5.5% vs. 10.5%), although our sampling strategy enabled us to invite many eligible individuals. To understand the reasons for the low response, a systematic nonresponse analysis using claims data from all participants and nonparticipants is currently being conducted. Second, the proportion of dropouts was higher than expected (56.8% (TG) and 48.9% (CG) vs. 25.0%), due to failed appointment scheduling when no participating dentists were available, moving to a nursing home, withdrawn informed consent, incorrect contact details and death. From March 2020, the outcome assessment was further impeded by the COVID−19 pandemic. However, the proportions of TG and CG participants whose outcomes were assessed were comparable and their baseline characteristics did not differ. Third, because of the low response and high proportion of dropouts, the numbers of TG and CG participants whose outcomes could be assessed was lower than expected. Even though more than 92 persons per group could be assessed, this limited the power of our analysis because the mean OHAT score in the TG was only half a point lower than in the CG and a difference of one OHAT point had been expected. In consequence, our subgroup analysis was underpowered even further. Fourth, the mean time between the provision of the intervention and the outcome assessment was longer than expected. Furthermore, the mean time between randomization and the outcome assessment was longer in the TG than in the CG. Both aspects might have attenuated the effectiveness of the intervention. This may also hold true for the fact that some of the assessed TG participants did not receive oral health education. However, this proportion was quite small. Fifth, the mean OHAT scores in our sample were lower than expected, suggesting that the participating home care recipients had better oral health than those who did not participate. Thus, our intervention might be more effective when implemented as part of the usual dental care. Sixth, the OHAT, OHIP and PSI comprise no detailed dental examination, but enabled us to assess oral health without creating any unnecessary burden for the participants. Seventh, due to the low number of participating dentists, it was not possible to assess the outcomes by more than one dentist per participant. Therefore, we were not able to assess interrater reliability. Eighth, because the OHAT, OHIP and PSI were only assessed at t_1_, we were unable to compare the outcomes between t_0_ and t_1_ for the TG. Furthermore, we were unable to compare the oral health status between the TG and CG at t_0_. However, due to the randomization, structural equivalence of all baseline characteristics between both groups can be assumed. Finally, only insurance funds belonging to the BKK Dachverband were selected which might have limited the representativeness and generalizability of our results.

## 5. Conclusions

The oral health intervention tended to improve objective oral health and reduced the prevalence of periodontal problems among home care recipients. Because the response to invitation letters sent out via insurance funds was poor and the impact of the intervention was low, future studies should investigate whether a pro-active recruitment strategy involving general practitioners and formal caregivers, combined with a more comprehensive intervention including dental treatment in the patients’ domicile would be more successful in improving oral health among home care recipients. Moreover, reforms are needed to attenuate existing barriers that currently impede the utilization and provision of dental care in the domestic setting.

## Figures and Tables

**Figure 1 ijerph-18-09339-f001:**
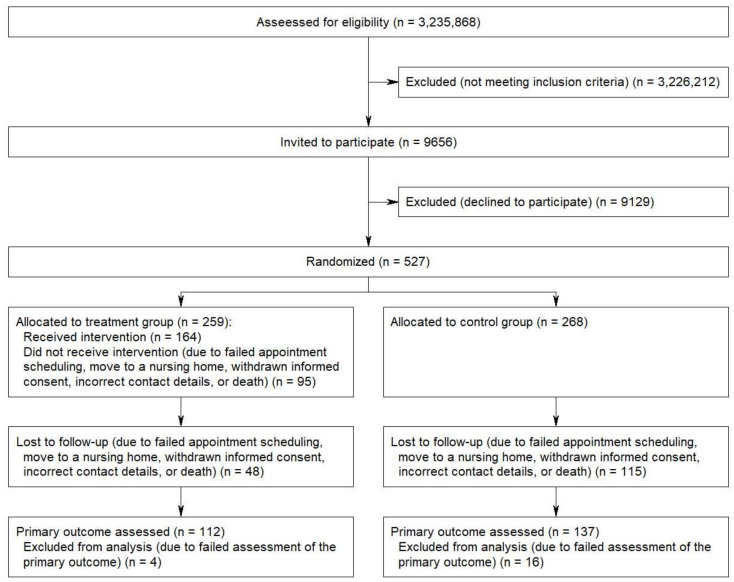
Flow diagram.

**Table 1 ijerph-18-09339-t001:** Characteristics of the treatment group and control group participants whose outcomes were assessed/non-assessed.

Category	Treatment Group (*n* = 259)			Control Group (*n* = 268)			*p*-Value Assessed in the Treatment Group vs. Assessed in the Control Group
	Primary Outcome Assessed (*n* = 112)	Primary Outcome Not Assessed (*n* = 147)	*p*-Value Assessed vs. Not Assessed	Primary Outcome Assessed (*n* = 137)	Primary Outcome Not Assessed (*n* = 131)	*p*-Value Assessed vs. Not Assessed	
	%	%		%	%		
**Sex**							
male	42.9	54.4		53.3	47.3		
female	57.1	45.6	0.0652	46.7	52.7	0.3296	0.1015
**Age group**							
<60 years	25.0	12.2		25.5	16.8		
60–74 years	25.0	17.0		24.1	16.0		
75–84 years	34.8	41.5		31.4	38.2		
85+ years	15.2	29.3	**0.0029**	19.0	29.0	**0.0406**	0.8580
mean (SD)	69.2 (18.4)	76.6 (14.5)	**0.0003**	68.6 (19.0)	74.4 (16.5)	**0.0036**	0.9647
**Long-term care grade**							
1/2	50.0	48.3		56.2	45.0		
3	26.8	31.3		29.2	27.5		
4/5	23.2	20.4	0.7015	14.6	27.5	**0.0297**	0.2179
**Type of LTC benefits**							
only informal care	76.8	72.1		74.5	74.0		
also formal care	23.2	27.9	0.3945	25.5	26.0	0.9393	0.6702

Note: Boldface indicates significant differences (*p* < 0.05). Abbreviation: SD, standard deviation.

**Table 2 ijerph-18-09339-t002:** Oral Health Assessment Tool (OHAT) scores for participants of the treatment group and participants of the control group.

Category	Treatment Group (*n* = 112)			Control Group (*n* = 137)			*p*-Value
	0 = Healthy	1 = Changes	2 = UnHealthy	0 = Healthy	1 = Changes	2 = Unhealthy	
	%	%	%	%	%	%	
Lips	90.2	8.0	1.8	83.8	14.7	1.5	0.2693
Tongue	79.1	18.2	2.7	75.9	21.2	2.9	0.8901
Gums and tissues	57.7	33.3	9.0	45.6	38.2	16.2	0.1015
Saliva	67.9	22.3	9.8	75.9	21.2	2.9	0.0651
Natural teeth ^a^	48.9	40.2	10.9	48.7	35.0	16.2	0.4867
Dentures ^b^	59.7	31.2	9.1	61.0	26.8	12.2	0.7311
Oral cleanliness	50.9	34.8	14.3	38.0	37.2	24.8	0.0534
Dental pain	92.9	7.1	0.0	88.2	8.8	2.9	0.1971

Notes: Missings (*n* = 1 (lips), *n* = 2 (tongue), *n* = 2 (gums and tissues), *n* = 2 (natural teeth), *n* = 3 (dentures) and *n* = 1 (dental pain)) were not considered. ^a^
*n* = 94 (treatment group); *n* = 117 (control group). ^b^
*n* = 78 (treatment group); *n* = 84 (control group).

**Table 3 ijerph-18-09339-t003:** Oral Health Impact Profile (OHIP) scores for participants of the treatment group and participants of the control group.

Dimension and Item	Treatment Group (*n* = 112)					Control Group (*n* = 137)					*p*-Value
	0 = Never	1 = Hardly Ever	2 = Occa-sionally	3 = Fairly Often	4 = Very Often	0 = Never	1 = Hardly Ever	2 = Occa-sionally	3 = Fairly Often	4 = Very Often	
	%	%	%	%	%	%	%	%	%	%	
**Functional limitation**											
Trouble pronouncing words	73.9	12.6	7.2	0.0	6.3	75.2	9.5	8.0	2.2	5.1	0.5243
Taste worse	75.2	10.1	3.7	6.4	4.6	75.2	5.8	10.9	1.5	6.6	**0.0389**
**Physical pain**											
Painful aching	69.6	10.7	8.9	8.0	2.7	61.3	9.5	15.3	6.6	7.3	0.2415
Uncomfortable to eat	56.3	10.7	16.1	10.7	6.3	66.4	7.3	7.3	7.3	11.7	0.0652
**Psychological discomfort**											
Self-conscious	67.0	8.9	12.5	8.0	3.6	70.8	8.8	8.0	5.1	7.3	0.4475
Tense	69.1	11.8	10.0	5.5	3.6	70.1	9.5	8.8	8.0	3.6	0.9081
**Physical disability**											
Diet unsatisfactory	76.6	10.8	4.5	3.6	4.5	83.9	6.6	1.5	4.4	3.6	0.4111
Interrupt meals	77.7	9.8	7.1	2.7	2.7	81.6	10.3	2.2	3.7	2.2	0.4473
**Psychological disability**											
Difficult to relax	66.7	9.0	13.5	6.3	4.5	70.6	11.0	5.1	8.1	5.1	0.2393
Been embarrassed	70.3	14.4	9.0	3.6	2.7	74.5	6.6	10.2	5.1	3.6	0.3511
**Social disability**											
Irritable with others	77.5	9.0	9.0	4.5	0.0	84.6	6.6	5.1	2.2	1.5	0.3309
Difficulty doing jobs	66.4	10.9	7.3	7.3	8.2	83.9	4.4	2.9	4.4	4.4	**0.0292**
**Handicap**											
Life unsatisfying	60.9	10.9	7.3	17.3	3.6	67.2	8.8	10.2	6.6	7.3	0.0646
Unable to function	71.2	10.8	6.3	6.3	5.4	86.1	4.4	4.4	2.9	2.2	0.0586

Notes: Boldface indicates significant differences (*p* < 0.05). Missings (*n* = 1 (trouble pronouncing words), *n* = 3 (taste worse), *n* = 2 (tense), *n* = 1 (diet unsatisfactory), *n* = 1 (interrupt meals), *n* = 2 (difficult to relax), *n* = 1 (been embarrassed), *n* = 2 (irritable with others), *n* = 2 (difficulty doing jobs), *n* = 2 (life unsatisfying) and *n* = 1 (unable to function)) were not considered.

## Data Availability

Due to data protection, data sharing is only feasible upon reasonable request and in collaboration with the authors.

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
