# Peer review of "Effectiveness of a Dental Intervention to Improve Oral Health among Home Care Recipients: A Randomized Controlled Trial"

_ijerph, 2021, doi:10.3390/ijerph18179339_

Round 1
Reviewer 1 Report
I think it's a very interesting topic for improving oral health through home care during the Corona era.
However, I give some statistically technical questions and some reviews to help the reader understand, so please check and answer those parts and reflect them in the revision version if possible. In fact, despite the great interest in the field and the apparent limitations of the research, I hope that relevant references and research will be steadily reported to develop LTC or the oral health program in the field.
Ln 81-83
It seems that a more detailed description of RANDOMIZATION is needed to help the reader understand.
Ln84-85 It would be appreciated if you could add a checklist for the CONSORT contents that the author said reflected in the text to the supplement file. (http://www.consort-statement.org/)
Ln86-118
In fact, it seems not possible to assume that all subjects received the same content of a technical intervention. It can be seen in reports on improved indicators when applying comprehensive content such as the subject's oral health. As a results, I think what should be judged on the improvement effect of oral health according to participating in the program in the broader sense of evaluation of the effectiveness of intervention, and stated the limit part of the research on this in the conclusion. It is recommended that the author choose a title for the evaluation of the effectiveness of the intervention or the evaluation of the effectiveness of the program, and please tell me the reason.
Ln122-124 / 148-153
If the examination for outcome assessment is carried out by one of the dentists or a nurse, the part about the expertise of both occupations is recognized.
However the examination during the calibration training process and the contents of reliability or validity should be presented in advance. Please describe additionally. If the evaluation of results for deriving academic results is limited to OHAT or if it has been involved in scoring such as PSR or PSI, additional descriptions are needed for that part.
Ln 159 I would appreciate if you could explain the difference between formal care and informal care.
Ln161-162
It is thought that additional explains are needed in the discussion section about the power limit that appeared because the group used for the final analysis did not reach the sample size calculated by the author. Or, please add a reasonable answer or description for this.
Ln 244
Please answer the reasons for providing information about groups that did not have a primary outcome. Excluding for no specific reason seems likely to reduce confusions.
It would be appreciated if you could additionally describe the record of the correction variables of the linear regression model presented in Table S1 provided by the supplement data.
TABLE 3
In the case of OHIP score, what do you think of the interpretation of the part where the OHIP score in the TG group was rather poorly evaluated?
Author Response
Response to Reviewer 1
Manuscript ID: ijerph-1317345
I think it's a very interesting topic for improving oral health through home care during the Corona era.
However, I give some statistically technical questions and some reviews to help the reader understand, so please check and answer those parts and reflect them in the revision version if possible. In fact, despite the great interest in the field and the apparent limitations of the research, I hope that relevant references and research will be steadily reported to develop LTC or the oral health program in the field.
- Ln 81-83
It seems that a more detailed description of RANDOMIZATION is needed to help the reader understand.
Authors’ response: Thank you very much for reviewing our manuscript and for your valuable comments. We now provide a more detailed description of the randomization process (page 2 in the revised manuscript with changes tracked).
- Ln84-85
It would be appreciated if you could add a checklist for the CONSORT contents that the author said reflected in the text to the supplement file. (http://www.consort-statement.org/)
Authors’ response: We added the CONSORT 2010 checklist to our supplementary material as suggested. Please note that the page numbers mentioned in the checklist refer to the page numbers in the revised manuscript with all changes accepted.
- Ln86-118
In fact, it seems not possible to assume that all subjects received the same content of a technical intervention. It can be seen in reports on improved indicators when applying comprehensive content such as the subject's oral health. As a results, I think what should be judged on the improvement effect of oral health according to participating in the program in the broader sense of evaluation of the effectiveness of intervention, and stated the limit part of the research on this in the conclusion. It is recommended that the author choose a title for the evaluation of the effectiveness of the intervention or the evaluation of the effectiveness of the program, and please tell me the reason.
Authors’ response: It is correct that not all subjects received the same intervention. The oral health assessment was provided to all participants of the treatment group, but dental treatment was of course only recommended if it was necessary (in 65.2% of all cases). Furthermore, the oral health education was tailored according to the demand of the respective patient. However, this is a typical feature of such complex interventions. A limitation is that we aimed to provide oral health education to all participants of the treatment group but 6.7% of the assessed treatment group participants did not receive the education which might have attenuated the effectiveness of the intervention. We now mention this limitation in our “Strengths and limitations” section (page 11, limitation 4). Furthermore, we now use the term “effectiveness” in our title and throughout the manuscript as suggested.
- Ln122-124 / 148-153
If the examination for outcome assessment is carried out by one of the dentists or a nurse, the part about the expertise of both occupations is recognized.
However the examination during the calibration training process and the contents of reliability or validity should be presented in advance. Please describe additionally. If the evaluation of results for deriving academic results is limited to OHAT or if it has been involved in scoring such as PSR or PSI, additional descriptions are needed for that part.
Authors’ response: We now provide more information on the training process of the participating dentists (page 3, “Intervention” section). Furthermore, we included a paragraph saying that in cases where a dentist was available, he/she assessed the primary and both secondary outcomes for a participant during one visit, whereas in cases where only a study nurse was available (5% of all outcome assessments), the secondary outcome II was not assessed to avoid adverse events (page 3, “Outcome assessment” section). OHAT and OHIP were validated by Chalmers et al. 2005 and John et al. 2006, respectively, whereas PSI/PSR is a widely used routine assessment applied in usual dental care (page 4). Due to the low number of participating dentists, it was not possible to assess the outcomes by more than one dentist per participant. Therefore, we were not able to assess interrater reliability. We included this important limitation into our “Strengths and limitations” section (page 11, limitation 7).
- Ln 159
I would appreciate if you could explain the difference between formal care and informal care.
Authors’ response: We added the requested explanation (page 4, “Record linkage” section).
- Ln161-162
It is thought that additional explains are needed in the discussion section about the power limit that appeared because the group used for the final analysis did not reach the sample size calculated by the author. Or, please add a reasonable answer or description for this.
Authors’ response: We did not reach the precalculated sample size and power because the response among home care recipients to the invitation letters was lower than expected and the proportion of dropouts was higher than expected. The dropouts were due to failed appointment scheduling when no participating dentists were available, moving to a nursing home, withdrawn informed consent, incorrect contact details, death, and the COVID-19 pandemic. We now state right at the beginning of our “discussion” section that we did not reach the precalculated sample size and power (page 9). Furthermore, we now discuss this aspect in our “Strengths and limitations” section in more detail (limitations 1 to 3, page 10 to 11).
- Ln 244
Please answer the reasons for providing information about groups that did not have a primary outcome. Excluding for no specific reason seems likely to reduce confusions.
Authors’ response: We provide this information to increase transparency regarding the high proportion of dropouts which might have biased our results. To avoid any confusion, we revised our “Statistical analysis” section (page 5) and the title and presentation of Table 1.
It would be appreciated if you could additionally describe the record of the correction variables of the linear regression model presented in Table S1 provided by the supplement data.
Authors’ response: Table S1 in the Supplementary Material presents the descriptive OHAT results for individuals receiving informal care only. The results of all 12 regression models (i.e., 4 of the main analysis presented in the main manuscript text and 8 of the subgroup analysis presented in the Supplementary Material) are only presented in text form. To avoid any confusion, we now reordered the presentation of our results throughout the Supplementary Material, beginning with the descriptive results. Furthermore, we added information telling the reader that the regression models mentioned in the Supplementary Material were controlled for sex, age group, LTC grade, and time in days between randomization and t1.
- TABLE 3
In the case of OHIP score, what do you think of the interpretation of the part where the OHIP score in the TG group was rather poorly evaluated?
Authors’ response: Regarding the subjective oral health assessed by OHIP, although not statistically significant, the total OHIP mean score as well as the percentage distribution of the total OHIP scores were poorer evaluated in the treatment group than in the control group. Looking at the individual OHIP items presented in Table 3, the item “difficulty doing jobs” was the only item rated significantly poorer in the treatment group compared to the control group. The tendency for a poorer subjective oral health at t1 in the treatment group might be explained by an increased awareness of personal oral health problems due to dentally assessed oral health problems during the provision of the intervention at t0. We included this interpretation into our “Discussion” section (page 10).
Additional note from the authors: In the revised version of our manuscript, we changed some numbers in the Abstract, “Results” section, and Supplementary Material. The changes result from a more precise definition of age group, long-term care grade, and type of LTC benefits affecting few individuals. We now define these variables for all participants in the second quarter of 2018 (page 4, “Record linkage” section). The interpretation of all results remained unchanged.
Reviewer 2 Report
The manuscript “Effects of a dental intervention to improve oral health among 2 home care recipients: a randomized controlled trial” reported a pragmatic intervention approach to improve oral health in older aged people. This is a very important contribution to the scientific literature, and I suggest some changes how the text and results are reported. There are some details of the outcomes that would benefit from further elaboration.
Line 48. To make this manuscript more understandable for the international readership I suggest explaining, which countries have been investigated in the reported references. Maybe also include references worldwide.
Line 70. Please briefly explain what “need of LTC” entails.
Line 88. Are the authors able to explain what “care as usual” entails?
Line 93 ff. This needs more explanations. What are the criteria to grad oral health? How does the dentist decided about treatment needs? Was this completely subjective?
Line 167 ff. “The data on LTC benefits were originally assessed for billing purposes by the insurance funds and indicate….” Can you please identify how valid this method of stratification is?
Line 209. Recruiting 5.5% of a eligible sample is not representative. It would be interesting to understand the reasons for participating and why most of the elderlies are not willing to participate.
Line 215. It would be helpful for the international readership to understand what different LTC grades mean and how they affect the outcomes of the study.
Line 231. Why did not the authors report the changes in the primary outcomes between t0 and t1 for the intervention group? This would help to understand the efficacy of the intervention and provide information, why the periodontitis status between t0 and t1 did not significantly change. Something I would expect because of dental treatment. Please discuss.
Table 1. I think the tables are hard to read, especially assessed/not assessed column. I acknowledge the intention of the authors to report that their outcomes are based only on a limited number of assessments, but would it be also possible to run a sensitivity analysis to investigate the effect that not all participants were assessed? The authors may also think about considering those participants who were not assessed as drop-outs and treat all data as intention-to-treat?
Line 266. No difference in PSI scores is highly surprising after dental treatment. Please explain potential reasons. Also, table 3 line “unable to function”, the outcomes are poorer (not significant) in the treatment group. I think this should be reported in the discussion as it provides important information for future studies.
Line 283. The authors selected “Betriebskrankenkassen”. Why? Does this affect the recruitment?
Line 295. Is an increase in enumeration the only way to increase the participation of dentists in their role as health care providers? Please discuss?
Discussion. The discussion is very much focussed on the German system. Accessibility, costs, poor oral health of our elderly is a general problem and the manuscript would benefit from a broader, worldwide perspective.
Author Response
Response to Reviewer 2
Manuscript ID: ijerph-1317345
The manuscript “Effects of a dental intervention to improve oral health among 2 home care recipients: a randomized controlled trial” reported a pragmatic intervention approach to improve oral health in older aged people. This is a very important contribution to the scientific literature, and I suggest some changes how the text and results are reported. There are some details of the outcomes that would benefit from further elaboration.
- Line 48. To make this manuscript more understandable for the international readership I suggest explaining, which countries have been investigated in the reported references. Maybe also include references worldwide.
Authors’ response: Thank you very much for your valuable time and helpful comments. According to this and your last comment, we now better indicate the origin of the literature cited in the “Introduction” (pages 1 to 2 in the revised manuscript with changes tracked) and “Discussion” sections (page 9 to 10). The selection of references was not limited to specific countries and included references worldwide.
- Line 70. Please briefly explain what “need of LTC” entails.
Authors’ response: We now explain that “need of LTC” means needing permanent support to compensate physical and/or mental disabilities (page 2).
- Line 88. Are the authors able to explain what “care as usual” entails?
Authors’ response: We now explain that, in Germany, usual dental care financed by the insurance funds includes, inter alia, dental prophylaxis, dental and periodontal treatment, and the supply of dentures (page 2, “Intervention” section).
- Line 93 ff. This needs more explanations. What are the criteria to grad oral health? How does the dentist decided about treatment needs? Was this completely subjective?
Authors’ response: The total oral health assessment provided at t0 to the treatment group participants was completely subjective in order to grade oral health and assess need for dental treatment and oral hygiene support. We added this information to the “Intervention” section (page 3).
- Line 167 ff. “The data on LTC benefits were originally assessed for billing purposes by the insurance funds and indicate….” Can you please identify how valid this method of stratification is?
Authors’ response: We used the administrative data on LTC benefits to differentiate between participants receiving only informal care and participants receiving also formal care. Because these data are originally generated for billing purposes and LTC benefits for formal care are much higher than for informal care, we assume that this method of stratification is highly valid.
- Line 209. Recruiting 5.5% of a eligible sample is not representative. It would be interesting to understand the reasons for participating and why most of the elderlies are not willing to participate.
Authors’ response: We agree that it is relevant to understand the reasons for the low response. In the further course of the study, we aim to conduct a systematic nonresponse analysis using claims data from the 527 participants and 9.129 nonparticipants. Because including this comprehensive analysis in this publication would have been beyond the scope of it and we have only recently received data on nonparticipants, these results will be part of a separate publication in the future. We included this information into our “Strengths and limitations” section (page 10, limitation 1).
- Line 215. It would be helpful for the international readership to understand what different LTC grades mean and how they affect the outcomes of the study.
Authors’ response: We now explain in the beginning of the “Statistical analysis” section what the different LTC grades mean (pages 4 to 5). Because the distribution of LTC grades was comparable in our treatment group and control group, we do not believe that LTC grades affected the results of our study.
- Line 231. Why did not the authors report the changes in the primary outcomes between t0 and t1 for the intervention group? This would help to understand the efficacy of the intervention and provide information, why the periodontitis status between t0 and t1 did not significantly change. Something I would expect because of dental treatment. Please discuss.
Authors’ response: When we planned our intervention study, we decided to assess OHAT, OHIP, and PSI only at t1 in both groups because an assessment at t0 in the control group would be unethical and would be some kind of intervention, too. Furthermore, we also wanted to evaluate the single effect of the intervention in the treatment group not including information relying on these further assessments. If we would have assessed OHAT, OHIP, and PSI in the treatment group also at t0, we would have evaluated a combination of the effects of the intervention as well as of a further assessment of the OHAT, OHIP, and PSI. We now describe this aspect in more detail in our “Outcome assessment” section (page 3) and added it to our “Strengths and limitations” section (page 11, limitation 8). Moreover, we now discuss that the non-significant change in the prevalence of periodontitis might be explained by the aspect that our intervention comprised no dental treatment, and it was the decision of the participants whether to follow the treatment recommendations or not (page 10, “Discussion” section).
- Table 1. I think the tables are hard to read, especially assessed/not assessed column. I acknowledge the intention of the authors to report that their outcomes are based only on a limited number of assessments, but would it be also possible to run a sensitivity analysis to investigate the effect that not all participants were assessed? The authors may also think about considering those participants who were not assessed as drop-outs and treat all data as intention-to-treat?
Authors’ response: We provide the detailed Table 1 to increase transparency regarding the high proportion of dropouts which might have biased our results. To make Table 1 easier to read, we removed the three footnotes and included them into the headings of the columns containing the p-values. Because the proportions of TG and CG participants whose outcomes were assessed were comparable and their baseline characteristics did not differ significantly, we decided to conduct only linear/logistic regressions to control for potential imbalance between the treatment group and control group.
- Line 266. No difference in PSI scores is highly surprising after dental treatment. Please explain potential reasons. Also, table 3 line “unable to function”, the outcomes are poorer (not significant) in the treatment group. I think this should be reported in the discussion as it provides important information for future studies.
Authors’ response: We extended our “Discussion” section and now discuss all aspects mentioned in this and your second last comment on line 231 (page 10).
- Line 283. The authors selected “Betriebskrankenkassen”. Why? Does this affect the recruitment?
Authors’ response: The idea for the project was developed in cooperation between the “BKK Dachverband”, the Universities of Bremen and Oldenburg, and the German Society for Gerodontology. Because of the involvement of the “BKK Dachverband”, which is an umbrella organization of “Betriebskrankenkassen” (which make up about 80 of 100 different statutory health and LTC insurance funds in Germany), only “Betriebskrankenkassen” were included, which might have limited the representativeness and generalizability of our results. We added the information that only insurance funds belonging to the BKK Dachverband were selected to our “Materials and Methods” section (page 2). Furthermore, we added the related limitation to our “Strengths and limitations” section (page 11, limitation 9).
- Line 295. Is an increase in enumeration the only way to increase the participation of dentists in their role as health care providers? Please discuss?
Authors’ response: We believe that, in addition to an increase in remuneration, dentists interested in geriatric dentistry should be equipped with mobile dental treatment facilities (e.g., mobile ultrasonic devises for removing tartar). Moreover, the implementation of centers for geriatric dentistry could help to ensure that an adequate number of dentists is available for the provision of dental care to LTC dependents. We extended our “Discussion” section accordingly (page 9).
- Discussion. The discussion is very much focussed on the German system. Accessibility, costs, poor oral health of our elderly is a general problem and the manuscript would benefit from a broader, worldwide perspective.
Authors’ response: We agree that the topic addressed in our study is of relevance worldwide. According to this and your first comment, we revised our “Discussion” section and now indicate the origin of the cited literature which includes many publications from all over the world (pages 9 to 10).
Additional note from the authors: In the revised version of our manuscript, we changed some numbers in the Abstract, “Results” section, and Supplementary Material. The changes result from a more precise definition of age group, long-term care grade, and type of LTC benefits affecting few individuals. We now define these variables for all participants in the second quarter of 2018 (page 4, “Record linkage” section). The interpretation of all results remained unchanged.
Round 2
Reviewer 2 Report
Thank you very much for addressing the comments.